# Impacts of Heifer Post-Weaning Intake Classification on Performance Measurements of Lactating and Non-Lactating Two-, Five-, and Eight-Year-Old Angus Beef Females

**DOI:** 10.3390/ani12131704

**Published:** 2022-06-30

**Authors:** Krista R. Wellnitz, Cory T. Parsons, Julia M. Dafoe, Darrin L. Boss, Samuel A. Wyffels, Timothy DelCurto, Megan L. Van Emon

**Affiliations:** 1Department of Animal and Range Sciences, Montana State University, Bozeman, MT 59717, USA; krista.eiseman@student.montana.edu (K.R.W.); samwyffels@montana.edu (S.A.W.); timothy.delcurto@montana.edu (T.D.); 2Northern Agricultural Research Center, Montana State University, Havre, MT 59501, USA; cory.parsons2@chsinc.com (C.T.P.); julia.dafoe@montana.edu (J.M.D.); dboss@montana.edu (D.L.B.)

**Keywords:** beef cattle, efficiency, heifer, intake, lactation

## Abstract

**Simple Summary:**

The total cost of production for cow-calf producers is increasing; with feed costs accounting for about 65% of total input costs per year. Development and selection of replacement beef heifers is a key factor in improving overall herd productivity and profitability. Selecting for replacement females with greater voluntary feed intake per unit body weight may lead to increased efficiency and productivity on low-quality, high-fiber forages typically seen in the Western United States. Higher voluntary intake may be related to greater rumen-reticulo volume as a proportion of the overall body volume. Therefore, selecting for animals with these attributes may provide greater efficiency and decreased input costs for Western beef producers in low-quality, forage-based systems.

**Abstract:**

Heifer post-weaning intake classification was utilized to evaluate subsequent intake and performance measurements of 2-, 5-, and 8-year-old lactating and non-lactating Angus females. For both studies, heifers were categorized based on voluntary feed intake (expressed as g/kg BW) as either low (<−0.50 SD from the mean), average (±0.50 SD from the mean), or high (>0.50 SD from the mean) within one year. Intake and production data of pregnant, non-lactating (*n* = 59; Study 1) and lactating, non-pregnant (*n* = 54; Study 2) females were evaluated. Heifer post-weaning voluntary feed intake was calculated over 80 test days post-weaning using GrowSafe feed intake units. Cow body-weight (BW) for non-lactating cows showed a tendency for age × intake interaction (*p* = 0.10), with older cows weighing more than younger cows. Milk production expressed as kilograms and g/kg BW of the cow had an age × intake (*p* < 0.001) effect. Two-year-old cows with low- and average-intake classifications had greater milk production (*p* < 0.001) and milk produced expressed as g/kg BW (*p* < 0.001) than 2-year-old cows with high-intake classifications. Additionally, 5-year-old cows with average and high-intake classifications had greater milk production (*p* < 0.001) and g/kg BW (*p* < 0.001) compared to 5-year-old cows classified as low-intake. In summary, heifer post-weaning intake classification had minor impacts on performance measurements in the three age classes of beef females at two different production levels.

## 1. Introduction

Beef heifer development from weaning to breeding is a multifaceted management scheme that takes into account multiple disciplinary subjects including, but not limited to, physiological age, nutrition, reproduction, genetics, and environment. Development and selection of replacement beef heifers is a key factor in improving overall herd productivity and profitability. The ability to utilize post-weaning feed intake information as a selection tool could improve overall herd productivity [1]. With modern technology, it is easier to collect accurate individual feed intake patterns over time. However, research related to heifer development and post-weaning intake and management is limited regarding the use of post-weaning intake behavior as a metric to predict lifetime productivity and longevity.

Recommendations for heifer development programs, indicate that heifers should be approximately 60 to 66% of their mature body weight (BW) by breeding [2]; while more recent data suggests that 50 to 55% of mature BW is sufficient [3,4]. In addition, the most common period to manipulate heifer growth and development is between weaning and the first breeding season [3]. Numerous researchers have determined that feed intake is a heritable trait in beef cattle [5,6,7,8,9,10], with heritability values in growing cattle ranging from 0.20 to 0.48, respectively.

Additionally, it is well understood that heifers that conceive early in their first breeding season are more likely to have greater lifetime productivity than those calving later in the season [11]. Day and Nogueira [12] also concluded that heifers calving earlier in the breeding season will wean calves with heavier weaning weights. In addition, heifers that calve earlier in the calving season tend to continue to calve earlier in future calving seasons, leading to improved productivity [13].

In relation to reproduction, nutrition plays a vital role in overall production system efficiency. Research by Wiltbank and coworkers [14] determined that an increased plane of nutrition following weaning resulted in earlier puberty and increased conception rates [13] compared to those at a low plane of nutrition. Buskirk et al. [15] reported that increased post-weaning average daily gain (ADG; 0.6 vs. 0.4 kg/d) was beneficial to milk production and overall productivity of traditionally weaned heifers.

Springman and colleagues [16] conducted a 3-year study utilizing Angus-based heifers where heifers developed in a drylot on a high-energy diet had greater ADG and were heavier prior to breeding and had greater BW and percent of mature BW pre-breeding than range- and corn-residue-developed heifers, but no differences were observed from heifers developed in a drylot on a low-energy diet. Interestingly, pregnant heifer initial and final BW, dry-matter intake (DMI), ADG, feed efficiency, and residual feed intake (RFI) were not different between heifer development systems.

Feed efficiency and intake research is not a new idea, but it is becoming more important as feed costs continue to rise and the cost of production increases. Selection of replacement beef females that consume less feed per kg of BW may decrease inputs and therefore decrease costs to the producer. However, there is a drastic shift in protein deposition towards fat deposition around the time of puberty [17]. These changes have the potential to impact how BW changes and intake shifts over time, therefore influencing differences between growing and mature animals at different stages of production.

Research evaluating heifer post-weaning intake classification and its impacts on lactating and non-lactating females is limited. Therefore, the objectives of these studies were to evaluate the relationship of heifer post-weaning intake classifications to subsequent performance of lactating and non-lactating cows at three different ages.

## 2. Materials and Methods

Data used in these studies were part of a larger project as described by Parsons et al. [18]. Utilization of animals for this study was approved by the Agricultural Animal Care and Use Committee at Montana State University (AACUC #2018-AA12). All studies were carried out at Northern Agricultural Research Center (NARC; Havre, MT, USA; 48.5500° N, 109.6841° W).

### 2.1. Heifer Intake Trials

Since 2010, all newly weaned beef females have been utilized in a heifer RFI trial. Following weaning, all beef females are placed in GrowSafe-enabled pens (GrowSafe DAQ 4000E; GrowSafe System Ltd., Airdrie, AB, Canada). Each year, calves are weaned from the dam while on pasture between mid-September and early October and the heifer calves remain on pasture for approximately 45 days. Once off pasture, heifers are placed on a forage-based diet for 60 to 85 d in order to determine post-weaning residual feed intake (RFI). All heifers are provided ad libitum access to water and forage-based diet composed of primarily corn silage (30.4%), grass hay (41.1%), and alfalfa (28.5%) on a dry-matter basis dependent on year and availability of ingredients. All diets are formulated to meet maintenance requirements for growing moderate-frame beef heifers (10.5% crude protein and 66.0% total digestible nutrients; [19]). Heifer BW was taken over two consecutive days at the beginning and end of the trial and every 28 d thereafter to record BW gain. Heifers were categorized based on voluntary feed intake (expressed as grams per kg BW) as either low (<−0.50 SD from the mean), average (±0.50 SD from the mean), or high (>0.50 SD from the mean) within the year for both studies. Individual heifer post-weaning intake information was then utilized to determine potential impacts on productivity of these females during a lactating or non-lactating period.

### 2.2. Lactating and Non-Lactating Intake Trials

Two intake studies were complete in order to evaluate the impact of heifer post-weaning intake at different ages and two stages of beef cattle production. In both studies, individual cow body-weights (BW) and body condition scores (BCS) were collected. Following collection of BW and BCS, cattle were placed in Growsafe-enabled pens for DMI intake analysis. For both studies (lactating vs. non-lactating), treatments were replicated in two pens, each pen containing 16 Growsafe intake feeding units. Prior to the beginning of the study, cows were provided a 7 d acclimation period followed by a 14 d DMI intake recording period. Of those 14 d, seven of the highest accuracy days were used to calculate average DMI per individual animal following the DMI study. In addition, intake behavior variables, including DMI (kg ∙ d^−1^ and kg BW^−1^), DMI rate (g ∙ min^−1^), time spent at the feeder (min ∙ d^−1^), and CV were collected. CV was calculated based on kg/d.

### 2.3. Study 1: Pregnant, Non-Lactating Cows

Fifty-nine pregnant, non-lactating black Angus females were used to evaluate the impacts of heifer post-weaning intake and feed efficiency. Cows were provided with a commercially available free-choice alfalfa/straw pellet (49.53% alfalfa hay and 49.52% straw) formulated to meet the nutrient requirements for pregnant, non-lactating cows. At the start of the trial, (14 d post-weaning), cows were categorized based on intake as heifers as either low (−0.50 SD from the mean), average (±0.50 SD from the mean), or high (>0.50 SD from the mean) within the year for 2-, 5- and 8-year-old old cows. At the same time, cows were dry-lotted to obtain consistent shrunk BW values.

### 2.4. Study 2: Non-Pregnant, Lactating Cows

Fifty-four non-pregnant, lactating black Angus females were used to evaluate the impacts of heifer post-weaning intake and feed efficiency. Cows were provided with a commercially available free-choice alfalfa/grass pellet (79.05% alfalfa hay and 20.00% corn) formulated to meet the nutrient requirements for lactating cows. Similar to study 1, cows were categorized based on intake as heifers as either low (<−0.50 SD from the mean), average (±0.50 SD from the mean), or high (>0.50 SD from the mean) within year for 2-, 5- and 8-year-old cows. However, only cows that calved in the first 42 d of the calving period were used in this study in order to focus the study on calves that were conceived via artificial insemination. On average, the intake study was carried out from day 40 to day 60 postpartum. Following the end of the DMI trial, a weigh–suckle–weigh trial was conducted to evaluate the impacts of heifer post-weaning intake classification and cow age on milk production, as described by Williams et al. [20]. Calves were weighed before an 8 h separation period and after nursing in order to obtain milk-production estimates.

## 3. Statistical Analysis

The data used in this manuscript are available in Appendix A. For both studies, the influence of heifer post-weaning intake classification and cow age on initial cow body condition score (BCS) and BW were analyzed using a generalized linear model in an ANOVA framework (car; [21]) including, intake classification, cow age, and the interactions of intake classification and cow age as fixed effects. In addition, the influence of intake classification and cow age on intake and intake behavior variables were analyzed using a generalized linear mixed model (lme4; [22]) in an ANOVA framework (car; [21]) including intake classification, cow age, and the interactions of intake classification and cow age as fixed effects, with individual cow and pen as random intercepts to account for the autocorrelation of repeated measurements of intake variables. Data were plotted and transformed if needed to satisfy assumptions of normality and homogeneity of variance. An individual animal was considered the experimental unit. An alpha ≤ 0.05 was considered significant. Tendencies were reported when significance was *p* ≤ 0.10. The Tukey method was used to separate means when alpha was <0.05 (emmeans; [23]). All statistical analyses were performed in R [24].

## 4. Results

### 4.1. Study 1: Pregnant, Non-Lactating Cows

The effects of heifer post-weaning intake classification on beef cow performance measurements for pregnant, non-lactating females are detailed in Table 1. Cow BW showed significance for age (*p* < 0.001) and intake (*p* = 0.03) with 2-year-old cows being lighter than 5- and 8-year-old cows. Moreover, high-intake 2-year-old cows had heavier BW than low-intake 2-year-olds, while average-intake 2-year-old cows were intermediate. In addition, 5-year-old low-intake cows had heavier BW than average-intake 5-year-old cows, while high-intake 5-year-old cows were intermediate. High-intake 8-year-old cows had heavier BW than average-intake 8-year-old cows, while low-intake 8-year-old cows were intermediate. Cow BCS displayed an age × intake interaction (*p* = 0.02) where average-intake 5- and 8-year-old cows tended to have lower BCS than low- and high-intake cows, with 2-year-old cows having no effect of intake classification on BCS (*p* = 0.22). Cow age had an effect on dry-matter intake (kg/d; *p* = 0.05) with all classes of 2-year-old cows consuming less than all classes of 5- and 8-year-old cows. There were no effects of cow age or intake classification observed for DMI (g/kg BW; *p* ≥ 0.48). Intake rate (g ∙ min^−1^) displayed an age effect (*p* < 0.01) with 5- and 8-year-old cows consuming feed faster than 2-year-old cows. Moreover, intake rate displayed a tendency for the cow age × intake interaction (g/min; *p* = 0.08) with high-intake 8-year-old cows consuming feed at a faster rate than low and average 8-year-old cows, while no differences were noted in 2- or 5-year-old cows. Cow CV displayed a tendency for an intake classification effect (*p* = 0.07) with high- and average-intake 2-year-old cows having greater CV than low-intake 2-year-old cows, while low-intake 5- and 8-year-old cows had greater CV than average- and high-intake 5- and 8-year-old cows. Time spent at the feeder displayed an age effect (*p* = 0.03) with 2-year-old cows spending more time at the feeder than 5- and 8-year-old cows. There was no effect of intake classification (*p* ≥ 0.22) for non-lactating females on cow BW, cow BCS, DMI (kg/d and g/kg BW), DMI rate (g/min), time spent at the feeder (min/d), or CV.

### 4.2. Study 2: Non-Pregnant, Lactating Cows

The effects of heifer post-weaning voluntary feed intake (g/kg BW) classification on subsequent beef cow performance, intake, and intake behavior for three age groups of non-pregnant, lactating females are detailed in Table 2. There was no effect of intake classification (*p* ≥ 0.39) for lactating females on DMI (expressed as kg/d or g/kg BW), DMI rate (g/min), coefficient of variation for intake, or time spent at the feeder (min/d). Cow BW displayed a cow age × intake interaction (*p* = 0.002), with low-intake 2-year-old cows having lighter BW than high-intake 2-year-old cows, while average-intake 2-year-old cows were intermediate. Alternatively, 5- and 8-year-old cows classified as low- or high-intake had increased BW when compared to average 5- and 8-year-old cows. Similarly, cow BCS displayed a cow age × intake interaction (*p* = 0.001), with low- and high-intake 5-year-old cows having greater BCS than average-intake 5-year-old cows, while no effect of intake classification were seen in 2- or 8-year-old cows. However, all classes of 2-year-old cows displayed a lower BCS than 5- and 8-year-old cows. Calf birth weights displayed an age × intake classification interaction (*p* ≤ 0.001) with offspring from low and average-intake 8-year-old cows having heavier birth weights than high-intake 8-year-old cows, while low-intake 5-year-old cows had greater calf BW than average- and high-intake 5-year-old cows. However, calf BW from 2-year-old cows did not differ between intake classifications. Calf Julian birth date displayed a cow age × intake interaction (*p* < 0.001) with high-intake 8-year-old cows calving later in the calving season than low- and average-intake 8-year-old cows, while low-intake 5-year-old cows calved earlier compared to average- and high-intake 5-year-old cows. In addition, high-intake 2-year-old cows calved earlier in the calving season compared to average- and low-intake cows. As expected, post-partum interval displayed an age effect (*p* < 0.001) with greater post-partum intervals for 2-year-old cows when compared to 5- and 8-year-old cows. Dry-matter intake (kg/d) displayed an age effect (*p* = 0.001) with 2-year-old cows consuming less than 5- and 8-year-old cows. Milk production (kg/d) displayed an age × intake effect (*p* < 0.001) with low-intake 2-year-old cows having greater daily milk production than average and high-intake 2-year-old cows, while average- and high-intake 5-year-old cows produced more than low-intake 5-year-olds. In addition, milk production did not differ between low-, average-, and high-intake 8-year-old cows. Similarly, milk production (g/kg BW) displayed an age × intake interaction (*p* < 0.001) with low-intake 2-year-old cows having greater daily milk production on a BW basis than average- and high-intake 2-year-old cows, while average- and high-intake 5-year-olds had greater milk production than low-intake 5-year-old cows. In addition, milk production on a BW basis did not differ between low-, average-, and high-intake 8-year-old cows.

## 5. Discussion

Development and selection of replacement beef heifers is a key factor in improving overall herd productivity and profitability. The ability to utilize post-weaning feed intake information as a selection tool for replacement heifers could improve overall herd productivity [1]. Increased longevity and efficiency in the herd provides economic advantages to the producer, where cows that remain in a herd longer reduce the cost of replacements on a yearly basis. The benefits from longer herd life come mainly from reduced costs of replacements and from increased income as a result of a higher proportion of cows producing at mature ages [25]. In addition, calves born from older cows have shown increased weaning weights and growth rates compared to calves from younger cows [26]. This difference is thought to be due to the increase in milk production with increasing age. Lubritz and colleagues, [27] stated that milk production in beef cows plateaus between 6 and 10 years of age. Similar to the current study, 6- and 8-year-old cows had greater milk production compared to 2-year-old cows.

Physiologically, we would also expect heifers that are larger in frame and capacity to have greater reticulo-rumen capacity than smaller-framed heifers; greater size leads to greater DMI [19]. Specifically, the capacity of the gut determines the capacity for digestion in herbivores [28]. The increase in reticulo-rumen volume allows for greater retention rates and digestion of forages that are typically high in fiber and low in quality. Ferrell and Jenkins [29] stated that a cow with genetic potential for high milk production will be more efficient when high-quality forages are available but fail to maintain milk production and body condition when low-quality forages are more prominent within their environment. In turn, the Western US and Northern Great Plains are nutritional environments generally characterized by low-quality high-fiber forages Presumably, these limited-nutrition environments would favor animals with greater reticulo-rumen volume and, as a result, greater forage intake expressed on a per unit of body weight basis. Furthermore, feed intake is going to be influenced by stage of production, environmental factors, and the genetic potential for overall size of the reticulo-rumen [30]. Research by Minson [31] reported that DMI in lactating cows fed primarily forages increases by 30% during the lactation stage. Parsons and colleagues [18] reported that cow BW of lactating and non-lactating females were greater in older cows than younger cows, however there were no differences in DMI intake (g ∙ kg BW^−1^). This was similar to the current results, where the only DMI difference was observed in 8-year-old cows for DMI rate.

Shike and coworkers [32] evaluated post-weaning heifer intake, RFI, and residual gain (RG) of Angus and Simmental × Angus heifers over a 5-year-period. Similar to the current study, heifers were classified as either low-, medium-, or high-intake and evaluated at 2 years of age. Results indicated that a greater percentage of females from the medium- or high-intake groups were retained in the herd compared to the low-intake heifers. This may be because later-born, lighter-framed heifers were at a disadvantage due to age and lack of time to grow. In addition, Shike et al. [32] also determined that cows that were classified as high-intake had calves that were heavier at birth than calves from cows categorized as either medium- or low-intake. Webster [33] stated that cows that consume more feed as a heifer are more likely to be larger as a mature animal, barring extreme nutritional or environmental stressors. Contrary to the current project, Shike and coworkers [32] did not observe an impact of heifer intake classification on milk production post-calving, while in the current study there was greater milk production in 2- and 5-year-old cows compared to 8-year-old cows. The data we observed in 2-year-old cows is not surprising as first-calf heifers are using greater amounts of energy than mature cows due to the initiation of lactation, continued skeletal growth and return to a functionally reproductive state for future breeding. While correlations between age of cow and milk production are low (0.32 and 0.22), milk yield has been observed to increase over the first three lactations [34]. In addition, RFI has been reported as moderately heritable at 0.45 [35]. Milk production of 5- and 8-year-old cows was greater compared to 2-year-old cows with an average of 3.4 kg, 4.51 kg, and 4.21 kg, respectively. In addition, these same young cows were considerably lighter compared to the 5- and 8-year-old cows. Lubritz and colleagues [27] reported that milk production in beef cow plateaus between 6 and 10 years of age, which may partly explain the differences we see from other studies. Similarly, Neville [36] reported that milk production increased linearly until age 6, while Rutledge et al. [37] reported up to 8 years of age.

In addition, there was no effect of intake classification for lactating females on DMI (kg/d and g/kg BW), DMI rate (g/min), coefficient of variation for intake, or time spent at the feeder (min/d). The current results suggest that heifer DMI classification may not have a long-lasting impact on lifetime productivity. Further research is needed in order to elucidate the differences in intake, BW gain, and milk production between the two different classes of beef cows classified over multiple age groups.

## 6. Conclusions

Results in this study suggest that heifer post-weaning intake classification did not greatly affect mature cow intake when expressed as g ∙ kg BW^−1^ for cows at three different ages and two different production stages—lactating and non-lactating. This information would indicate that utilization of post-weaning heifer intake classifications beginning about 45 d post-weaning may not be a strong metric for measuring the impact of lifetime productivity. This data tend to suggest that cow age has a greater impact on post-weaning intake than intake classification as a heifer. Further research is necessary to determine the ideal time to collect post-weaning intake data in order to accurately determine its overall impact on subsequent productivity in the beef herd.

## Figures and Tables

**Table 1 animals-12-01704-t001:** The influence of heifer post-weaning intake expressed as grams per kg of body weight on subsequent beef cow performance of three age classes of pregnant, non-lactating beef cows (Study 1).

	Cow Age, Years												
	2			5			8				*p*-Value		
Category	Low Intake	Average Intake	High Intake	Low Intake	Average Intake	High Intake	Low Intake	Average Intake	High Intake	SE ^1^	Age	Intake	Age × Intake
Cows, *n*	7	5	8	6	7	7	8	4	7				
Cow BW, kg ^2^	431.68 ^a^	433.63 ^ab^	540.35 ^b^	594.96 ^a^	570.55 ^b^	587.40 ^ab^	562.45 ^ab^	544.31 ^a^	587.40 ^b^	5.81	<0.001	0.03	0.10
Cow BCS ^3^	5.50	5.40	5.34	5.71 ^a^	5.28 ^b^	5.64 ^a^	5.41 ^a^	5.06 ^b^	5.39 ^a^	0.07	0.01	0.22	0.02
DMI, kg/d	12.82	12.37	12.84	15.46	16.65	16.13	15.43	17.77	17.19	1.38	0.05	0.91	0.69
DMI, g/kg BW	29.32	28.67	28.69	26.41	29.10	27.28	28.08	32.84	29.86	2.27	0.48	0.95	0.70
DMI, g/min	89.95	90.96	95.50	147.54	146.90	141.91	132.24 ^ab^	106.99 ^a^	150.57 ^b^	12.29	<0.01	0.87	0.08
CV, % ^4^	12.99	20.99	24.57	22.16	20.28	18.04	21.71	18.88	14.92	4.11	0.12	0.07	0.12
Time at feeder, min/d	149.46	140.97	138.08	108.06	115.49	118.32	127.06 ^ab^	171.98 ^a^	114.71 ^b^	10.51	0.03	0.72	0.06

^1^ Pooled standard error of the means, ^2^ cow body-weight (kg) at initiation of trial, ^3^ cow body condition score at initiation of trial, ^4^ coefficient of variation for intake expressed as kg ∙ d^−1^, ^a,b^ means within row and cow age lacking common superscript differ (*p* < 0.05), ^a,b^ heifers were categorized as low (>−0.50 SD from the mean), average (±0.50 SD from mean), or high (<+0.50 SD from the mean) intake classes.

**Table 2 animals-12-01704-t002:** The influence of heifer post-weaning intake expressed as grams per kg of BW on subsequent beef cow performance of three age classes of non-pregnant, lactating beef cows (Study 2).

	Cow Age, Years												
	2			5			8				*p*-Value		
Category	Low Intake	Average Intake	High Intake	Low Intake	Average Intake	High Intake	Low Intake	Average Intake	High Intake	SE ^1^	Age	Intake	Age × Intake
Cows, *n*	5	6	7	6	6	6	8	3	7				
Cow BW, kg ^2^	386.28 ^a^	399.46 ^ab^	417.82 ^b^	561.32 ^a^	556.40 ^b^	565.85 ^a^	530.42 ^a^	492.90 ^b^	533.94 ^a^	6.25	<0.001	0.001	0.002
Cow BCS ^3^	4.30	4.21	4.14	4.92 ^a^	4.63 ^b^	5.04 ^a^	4.66	4.58	4.57	0.06	<0.001	0.19	0.001
Calf, *n*	5	5 ^x^	7	6	6	6	8	3	7				
Calf BW, kg ^4^	93.78	94.26	94.61	102.96 ^a^	94.72 ^b^	92.23 ^b^	109.83 ^a^	112.19 ^a^	92.14 ^b^	1.77	<0.001	0.95	<0.001
Calf Julian birth date	69.40	69.83	66.71	68.83 ^a^	77.67 ^b^	76.50 ^b^	70.38 ^a^	72.33 ^a^	84.86 ^b^	1.63	0.75	0.29	<0.001
Post-partum interval, d	69.60	69.17	72.29	70.17 ^a^	61.33 ^b^	62.50 ^b^	68.63 ^a^	66.67 ^a^	54.14 ^b^	1.63	<0.001	0.75	0.29
DMI, kg/d	17.99	17.41	19.36	23.71	23.33	23.49	22.79	23.78	24.07	1.37	0.001	0.39	0.85
DMI, g/kg BW	45.81	43.58	46.43	43.10	41.97	41.08	44.24	48.79	44.44	2.65	0.73	0.64	0.58
DMI, g/min	127.65	115.52	133.54	165.10	152.54	174.94	152.77 ^ab^	123.56 ^a^	198.44 ^b^	13.87	0.13	0.56	0.21
CV, % ^5^	12.85	13.87	12.59	11.51	11.35	8.01	12.97	14.52	10.12	1.85	0.81	0.89	0.79
Time @ feeder, min/d	146.53	155.27	151.50	147.50	158.11	136.17	157.81 ^ab^	195.09 ^a^	125.15 ^b^	14.54	0.73	0.89	0.13
Milk production, kg/d	4.08 ^a^	3.54 ^a^	2.59 ^b^	3.86 ^a^	4.84 ^b^	4.84 ^b^	4.42	4.08	4.15	0.20	0.07	<0.001	<0.001
Milk production, g/kg BW	10.53 ^a^	8.87 ^a^	6.18 ^b^	6.94 ^a^	8.86 ^b^	8.63 ^b^	8.40	8.45	7.93	0.47	<0.001	<0.001	<0.001

^1^ Pooled standard error of the means, ^2^ cow body-weight (kg) at initiation of trial, ^3^ cow body condition score at initiation of trial, ^4^ calf body weight (kg) at weigh–suckle–weigh, ^5^ coefficient of variation for intake expressed as kg ∙ d^−1^, ^x^ one calf died and was removed from the dataset, ^a,b^ means within row and cow age lacking common superscript differ (*p* < 0.05), ^a,b^ heifers were categorized as low (>−0.50 SD from the mean), average (±0.50 SD from mean), or high (<+0.50 SD from the mean) intake classes.

## Data Availability

The data used in this manuscript are available in Appendix A.

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
