# Peer review of "Impacts of Heifer Post-Weaning Intake Classification on Performance Measurements of Lactating and Non-Lactating Two-, Five-, and Eight-Year-Old Angus Beef Females"

_animals, 2022, doi:10.3390/ani12131704_

Round 1
Reviewer 1 Report
Dear Authors,
The research that you present contributes to improve decision-making in production units, so it has a direct impact. After reviewing the manuscript, some doubts and comments arose.
Introduction
Line 56: Reference numbers [5-10] should be place at the end of the sentence.
Line 57: Why “respectively” if you only mention that “feed intake is a heritable trait in beef cattle”?
Materials and methods
Line 96: I suggest to include the date instead “to the present”.
Line 96: Indicate how many calves did you use?
Line 99: Mention the age of the heifers at weaning.
Line 126: Indicate pregnancy time.
Discussion:
Lines 272-286: It is necessary to discuss more about the interaction effect of post-weaning intake and age on milk production, since only the effect of age is discussed. According to the results, at 2 years of age, cows produce less milk when post-weaning intake was high, but the opposite effect occurs at 5 years of age.
Conclusion
Conclude taking into account all the results obtained, not only the intake.
Regards
Author Response
The research that you present contributes to improve decision-making in production units, so it has a direct impact. After reviewing the manuscript, some doubts and comments arose.
Introduction
Line 56: Reference numbers [5-10] should be place at the end of the sentence.
Author response: changed as suggested
Line 57: Why “respectively” if you only mention that “feed intake is a heritable trait in beef cattle”?
Author response: changed as suggested
Materials and methods
Line 96: I suggest to include the date instead “to the present”.
Author response: not changed
Line 96: Indicate how many calves did you use?
Author response: As the study only focuses on a small segment of the total, we do not have current numbers through 2022
Line 99: Mention the age of the heifers at weaning.
Author response: added as suggested
Line 126: Indicate pregnancy time.
Author response: added as suggested
Discussion:
Lines 272-286: It is necessary to discuss more about the interaction effect of post-weaning intake and age on milk production, since only the effect of age is discussed. According to the results, at 2 years of age, cows produce less milk when post-weaning intake was high, but the opposite effect occurs at 5 years of age.
Author response: additional discussion regarding intake classification was added
Conclusion
Conclude taking into account all the results obtained, not only the intake.
Author response: additional information was included in the conclusion
Reviewer 2 Report
Title: Impacts of heifer post-weaning intake classification on performance measurements of lactating and non-lactating two-, five-, and eight-year-old Angus beef females
This study gave an implication of the effect of heifer post-weaning intake classification on late cow performance, which is helpful for breed selection. Some specific points need to be considered:
L96-100 What’s the weaning time? And the age to pasture?
L100-102 More details are required on RFI determination. Were the cows group grazing, or individual grazing? Some description on pasture?
L106-108 Which is the trial beginning date, from weaning or mid-September on pasture?
L123-124 How to measure the behavior parameters?
Author Response
This study gave an implication of the effect of heifer post-weaning intake classification on late cow performance, which is helpful for breed selection. Some specific points need to be considered:
L96-100 What’s the weaning time? And the age to pasture?
Author response: added as suggested
L100-102 More details are required on RFI determination. Were the cows group grazing, or individual grazing? Some description on pasture?
Author response: RFI was determined in the GrowSafe pens for individual intake, explained in lines 100-104
L106-108 Which is the trial beginning date, from weaning or mid-September on pasture?
Author response: As stated in lines 99-104 – heifers were placed in Growsafe units 45d post-weaning and provided a forage-based diet for 60-85 d in order to determine RFI.
L123-124 How to measure the behavior parameters?
Author response: As indicated in Line 120, all heifers were placed in Growsafe units that measure individual animal intake and intake behavior
Reviewer 3 Report
The manuscript is well written and structured. The presentation of results is appropriate and in the discussion the relevant literature has been cited. There are just some minor points which may be changed.
Line 106 – 112 I do miss the heifer postweaning information, please include.
On what basis did you decide to choose the categorisation into feed intake classes < -0.05; ±0.05 and >0.5 SD?? Please explain
Line 116 BCS…please give a reference for BCS method…
Author Response
The manuscript is well written and structured. The presentation of results is appropriate and in the discussion the relevant literature has been cited. There are just some minor points which may be changed.
Line 106 – 112 I do miss the heifer postweaning information, please include
Author response: Heifer postweaning protocol is explained in lines 96-115
On what basis did you decide to choose the categorisation into feed intake classes < -0.05; ±0.05 and >0.5 SD?? Please explain
Author response: The authors don’t feel that this needs to be included in the paper, however, 0.5 SD was selected to provide data that indicate significant results (power) while providing relatively equal group numbers. In addition, it allows for replication if others choose to do that.
Line 116 BCS…please give a reference for BCS method…
Author response: further information provided as suggested
Reviewer 4 Report
Impacts of post-weaning intake classification …
This is a useful study of the performance of Angus beef cows based on their classification of intake after weaning. Classification is after weaning, being taken off pasture, on a forage diet. Score is on voluntary feed intake g/kg BW, expressed as deviations from the mean, in three groups; low, average or high.
The results have some significant values, but these are generally not consistent with each other. In some cases, the Average Intake group is more extreme than either the low or high intake. Or the trend for one age group is not seen in the other age groups. This is particularly the case in the lactating study, where milk production showed highly significant effects (p < 0.001), but these effects were not consistent across age groups. This may be a result of low numbers of animals in each group (3 to 8), but it is not obvious that a much larger number should have been used.
This does not mean the study is incorrect, or poorly done, but does require caution in the interpretation of results. The Discussion appears to cover the results and compares them with other work in this field and is well done.
However, the Simple Summary bears no relationship to the results obtained. It is brief summary of the introduction, where there are certain expectations about the results that might be obtained> however, it does not represent the results and discussion of this paper. It should be completely rewritten.
Minor comments
In line 109-110 and 131-132 the low and average groups are based on mean – 0.05 SD, but the high group is mean + 0.5 SD. Should these all be 0.5 SD, not 0.05 SD.
Line 114. completed, not complete
Line 121. “seven of the highest accuracy days were used”. What does this mean?
Table 2, subscripts. “Means within row and cow age lacking common superscripts differ (p < 0.05)”. But there are some results that do not have any subscript and therefore must differ because they do not have a common superscript. Should this be “Means within rows and cow age with different superscripts differ p < 0.05)”.
Author Response
Minor comments
In line 109-110 and 131-132 the low and average groups are based on mean – 0.05 SD, but the high group is mean + 0.5 SD. Should these all be 0.5 SD, not 0.05 SD.
Author response: modified to be consistent
Line 114. completed, not complete
Author response: modified as suggested
Line 121. “seven of the highest accuracy days were used”. What does this mean?
Author response: additional information added for clarity
Table 2, subscripts. “Means within row and cow age lacking common superscripts differ (p < 0.05)”. But there are some results that do not have any subscript and therefore must differ because they do not have a common superscript. Should this be “Means within rows and cow age with different superscripts differ p < 0.05)”.
Author response: changed as suggested